# Coupled Large Deformation Finite Element Formulations for the Dynamics of Unsaturated Soil and Their Application

**Nadarajah Ravichandran** [1,*] and **Tharshikka Vickneswaran** [2]

1   Glenn Department of Civil Engineering, Clemson University, 202A Lowry Hall, Clemson, SC 29634, USA
2   Civil and Environmental Engineering, University of Louisville, 218 Eastern Pkwy, Louisville, KY 40208, USA
*   Correspondence: nravic@clemson.edu; Tel.: +864-656-2818

**Abstract:** Unsaturated soil is a three-phase medium with three interfaces, and the mathematical equations that represent its behavior must be developed in a fully coupled manner for accurately predicting its hydromechanical behavior. In this paper, a set of fully coupled governing equations was developed for the dynamics of unsaturated soil, considering the interaction among the bulk phases and interfaces. In addition to implementing the complete governing equations, a simplified formulation was developed for practical applications. The derivation of the finite element formulation considering all the terms in the partial differential equations resulted in a formulation called *complete formulation* and was solved for the first time in this paper. Another formulation called *reduced formulation* was derived by neglecting the relative accelerations and velocities of water and air in the governing equations. In addition, small and large deformation theories were developed and implemented for both formulations. To show the applicability of the proposed models, the dynamic behavior of an unsaturated soil embankment was simulated using both small and large deformation formulations by applying minor and severe earthquakes. The reduced formulation was found to be computationally efficient and numerically stable. The smaller displacements predicted by large deformation theories show that the results are consistent with the expected behavior. Large deformation theories are considered suitable when the geotechnical system undergoes large deformation and may lead to accurate prediction.

**Keywords:** dynamics of unsaturated soil; fully coupled analysis; large deformation analysis; finite element framework

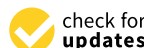



## 1. Introduction

The hydromechanical behavior of geotechnical structures, such as earth dams, levees, and near-surface soil that supports superstructures, is influenced by the degree of saturation (DOS) of the soil. The DOS of soil varies with climatic and hydrological parameters. In general, the soil is in an unsaturated state, and the saturated (DOS = 100%) and dry (DOS = 0) states are special cases of an unsaturated state. During hydrological and/or climatic events, such as rainfall, flood, and drought, the state of the soil changes continuously, resulting in variations in the behavior of geotechnical systems. To better design and construct sustainable and resilient geotechnical systems, the variation in the behavior of the soil when it transitions from one state to the other must be understood. In this study, numerical models that predict the hydromechanical behavior of soil are developed, considering the soil as unsaturated soil, and implemented.

Unsaturated soil consists of three bulk phases (solid, water, and air) and three interfaces (solid–water, water–air, and air–solid). The hydromechanical behavior of the soil is influenced by the bulk phases and interfaces and also by the interaction among them. Of the three interfaces, the water–air interface (contractile skin) has a significant influence on the hydromechanical behavior of unsaturated soil [1], and it must be incorporated at the governing equation level for to accurately predicting the behavior of soil. The behavior

of the water–air interface is controlled by the amount of water in the unsaturated soil, particularly at the particle contacts. The contractile skin helps maintain the pressure difference between the water and air phases and results in the water pressure in an unsaturated soil system being always negative. These factors complicate studying unsaturated soil compared to fully saturated and/or dry soils.

The partial differential equations for the dynamics of unsaturated soil systems can be derived based on physical laws, such as the balance of mass, linear momentum, angular momentum, and the first and second laws of thermodynamics. When developing governing equations, the macroscopic quantities representing microscopic quantities, such as the area density of the water–air interface, must be identified and properly used in the derivation. A rigorous volume-averaging technique is used to accomplish this task [2].

The partial differential equations are typically simplified to develop numerically stable formulations without compromising the accuracy of the prediction. In the case of unsaturated soils, the relative accelerations of the water and air phases with respect to the solid phase are typically neglected in the solution procedure [3,4]. However, such an assumption has not been proved to be valid. Schrefler et al. [3] and Wei [5] derived finite element formulations by neglecting the relative accelerations of the water and air phases. They solved the finite element equations, considering solid displacement ($\mathbf{u}$), water pressure ($p^l$), and air pressure ($p^g$) as the primary nodal unknowns ($\mathbf{u}\, p^l\, p^g$ formulation). In the formulation, they used a two-dimensional, four-node quadrilateral element with continuous bilinear displacement ($\mathbf{u}$) and pressure ($p^l$ and $p^g$). However, it is shown that such formulation violates the Babuska–Brezzi conditions or the simpler Zienkiewicz–Taylor patch test [6] for solvability and convergence [7].

In general, small deformation theories are commonly used in the finite element simulation of soils, assuming that the geotechnical engineering structures experience small deformations. The true response of the geotechnical structures undergoing large deformations cannot be predicted correctly with small deformation theories. Therefore, large deformation theories must be incorporated while solving the partial differential equations using the finite element method [8,9]. Sanavia et al. [10] and Gawin and Sanavia [11] developed a formulation for partially saturated soils undergoing large deformations. From the example simulations, they showed strain and negative pore pressure localization and their effects on the hydromechanical behavior of partially saturated soils.

In this paper, a set of fully coupled partial differential equations for the dynamics of unsaturated soils was derived, considering the interaction among the bulk phases and interfaces. Next, two different finite element formulations (complete and reduced) of the governing equations were derived for analyzing unsaturated soil mechanics problems undergoing large deformations. In addition, small and large deformation theories were developed and implemented for both formulations. Finally, the dynamic behavior of an unsaturated soil embankment was simulated using both small and large deformation formulations by applying minor and severe earthquake acceleration–time histories.

## 2. Summary of Governing Equations

The partial differential equations for the dynamics of unsaturated soils were derived based on the laws of physics, such as the momentum balance, mass balance, energy balance, and the second law of thermodynamics. Since the amount of water in the soil system is directly related to the matric suction (air pressure minus the water pressure), a constitutive equation that relates the amount of water to the matric suction was established. In this study, the volume fraction of the water phase is assumed to be a function of the solid skeleton's matric suction and volumetric strain, as shown in Equation (1).

$$n^l = n^l(S, \varepsilon_v) \tag{1}$$

where $n^l$ is the volume fraction of the water phase, $\varepsilon_v$ is the volumetric strain of the solid skeleton, and $S$ is the matric suction defined by $S = p^g - p^l$, where $p^g$ is the pore air pressure and $p^l$ is the pore water pressure.

### 2.1. Mass Balance Equations

When deriving the governing equations, the volume spanned by the solid phase is considered as the representative elementary volume (REV) and its motion is given by $\varphi^s(\mathbf{X}, t)$, where $\mathbf{X}$ is the material coordinate and $t$ is the time. The water and air phases can move in (influx) and out (outflux) of the REV. This phenomenon is graphically shown in Figure 1. Therefore, there will be net flow across the closed REV. Such influx and outflux were considered in this study when deriving the governing equations.

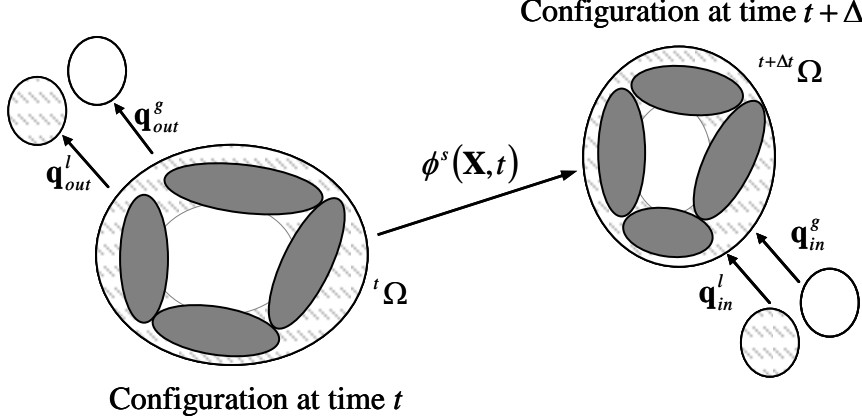

**Figure 1.** Motion of an REV and fluids in an unsaturated soil system.

The balance of mass in an REV for a bulk phase $\alpha$ is given by Equation (2).

$$m^\alpha(\Omega) = \int_{t+\Delta t\Omega} (n^\alpha \rho^\alpha)\, d\Omega \tag{2}$$

where $m^\alpha$ is the mass of the $\alpha$-phase, $n^\alpha$ is the volume fraction of the $\alpha$-phase, and $\rho^\alpha$ is the density of the $\alpha$-phase.

- Mass balance for the solid phase:

The following equation (Equation (3)) for the solid phase was derived, assuming that the solid particles are incompressible:

$$-\dot{n} + (1 - n)\,\mathrm{div}(\mathbf{v}^s) = 0 \tag{3}$$

where $n$ is the porosity of the unsaturated soil system and $\mathbf{v}^s$ is the velocity vector of the solid phase.

- Mass balance for the water phase:

The final mass balance equation for the water phase was derived by incorporating the mass balance equation for the solid phase into the mass balance equation for the water phase, as shown in Equation (4).

$$\left(\frac{\partial n^l}{\partial \varepsilon_v}\right)\dot{u}_{i,i} + n^l \dot{U}^l_{i,i} + \left(\frac{n^l}{\Gamma^l} - \frac{\partial n^l}{\partial S}\right)\dot{p}^l + \left(\frac{\partial n^l}{\partial S}\right)\dot{p}^g = 0 \tag{4}$$

where $u$ is the displacement of the solid phase, $U^l$ is the displacement of the water phase, and $\Gamma^l$ is the bulk modulus of the water phase.

- Mass balance for the air phase:

Similarly, the final mass balance for the air phase was derived by incorporating the mass balance equation for the solid phase into the mass balance equation for the air phase, as shown in Equation (5).

$$\left(1 - n - \frac{\partial n^l}{\partial \varepsilon_v}\right)\dot{u}_{i,i} + n^g\,\dot{U}^g_{i,i} + \left(\frac{\partial n^l}{\partial S}\right)\dot{p}^l + \left(\frac{n^g}{\Gamma^g} - \frac{\partial n^l}{\partial S}\right)\dot{p}^g = 0 \tag{5}$$

where $U^g$ is the displacement of the air phase and $\Gamma^g$ is the bulk modulus of the air phase.

### 2.2. Momentum Balance Equations

Momentum balance equations can fully describe the motion of the unsaturated soil system for the soil (solid, water, and air mixture), water phase, and air phase. The momentum balance equations for the water and air phases are essentially the generalized Darcy's flow equations in the flow direction. At the micromechanical level, the primary resistance to the flow of these fluids is the drag forces on the solid skeleton and the primary driving force is the fluid pressure gradients. The final momentum balance equations for soil, water, and air were derived, as shown in Equations (6)–(8), respectively.

- Linear momentum balance for the mixture:

$$n^s\,\rho^s\,\ddot{u}_j + n^l\rho^l\,\ddot{U}^l_j + n^g\rho^g\ddot{U}^g_j - \sigma_{ij,i} - \rho b_j = 0 \tag{6}$$

- Linear momentum balance for the water:

$$\rho^l\,\ddot{U}^l_j - \left(\hat{k}^l_{ij}\,n^l\right)\dot{u}_i + \left(\hat{k}^l_{ij}\,n^l\right)\dot{U}^l_i + \left(\delta_{ij}p^l\right)_{,i} - \rho^l b_j = 0 \tag{7}$$

- Linear momentum balance for the air:

$$\rho^g\ddot{U}^g_j - \left(\hat{k}^g_{ij}\,n^g\right)\dot{u}_i + \left(\hat{k}^g_{ij}\,n^g\right)\dot{U}^g_i + \left(\delta_{ij}p^g\right)_{,i} - \rho^g b_j = 0 \tag{8}$$

where $\sigma_{ij}$ is the total stress tensor, $b_j$ is the gravitational acceis the gravitational acceleration vector, leration vector, $\hat{k}^l_{ij}$ is the inverted permeability tensor of the water phase, $\hat{k}^g_{ij}$ is the inverted permeability tensor of the air phase, and $\delta_{ij}$ is the Kronecker delta.

## 3. Updated Lagrangian Formulation of Governing Equations

### 3.1. Application of the Principle of Virtual Work

The principle of virtual work requires that the virtual work performed when the soil body undergoes a virtual displacement $\delta\mathbf{u}$ be equal to the external work performed by the body force and traction, i.e.,

$$^{t+\Delta t}W^{\text{int}} = \int_{t+\Delta t\Omega} {}^{t+\Delta t}\sigma_{ij}\delta^{t+\Delta t}e_{ij}{}^{t+\Delta t}d\Omega \tag{9}$$

$$\begin{aligned}
^{t+\Delta t}W^{ext} &= \int_{t+\Delta t\Omega} {}^{t+\Delta t}\rho^{t+\Delta t}\,b_i\delta u_i{}^{t+\Delta t}d\Omega - \int_{t+\Delta t\Omega} {}^{t+\Delta t}n^{st+\Delta t}\rho^{s\,t+\Delta t}\ddot{u}_i\delta u_i{}^{t+\Delta t}d\Omega \\
&\quad - \int_{t+\Delta t\Omega} {}^{t+\Delta t}n^{lt+\Delta t}\rho^{l\,t+\Delta t}\ddot{U}^l_i\delta u_i{}^{t+\Delta t}d\Omega - \int_{t+\Delta t\Omega} {}^{t+\Delta t}n^{gt+\Delta t}\rho^{g\,t+\Delta t}\ddot{U}^g_i\delta u_i{}^{t+\Delta t}d\Omega \\
&\quad + \int_{t+\Delta t S^T} {}^{t+\Delta t}f^t_i\delta u_i{}^{t+\Delta t}dS
\end{aligned} \tag{10}$$

where ${}^{t+\Delta t}\sigma_{ij}$ are the Cartesian components of the Cauchy total stress tensor and ${}^{t+\Delta t}e_{ij}$ are the Cartesian components of the strain tensor. In the case of unsaturated soils, the body force, the inertial force of the solid skeleton, the inertial forces of the pore fluids, and the surface traction contribute to the external work. Similar equations were derived for the

motion of the pore fluids. It should be noted that there are two major difficulties that exist when applying these equations for developing large deformation theories, which involve rotation and a change in configuration. First, the configuration at time $t + \Delta t$ is unknown, and the integrals over the volume $^{t+\Delta t}\Omega$ and surface $^{t+\Delta t}S$ cannot be evaluated before calculating the equilibrium position at time $t + \Delta t$. The second difficulty is the presence of total stress in the internal work equation. Since the total stress does not directly influence the mechanical behavior of the soil, the principle of net stress, which is expressed in terms of Cauchy stress, was applied.

### 3.2. Stress–Strain Relationship for the Solid Skeleton undergoing Large Deformation

For the large deformation formulation, the rotation of the element must be separated from its deformation to accurately calculate the strain. During the elastoplastic deformation of the body from the reference configuration to the current configuration, the material undergoes elastic reversible deformation and plastic irreversible deformation. The motion of the body from $^{t}\Omega$ to $^{t+\Delta t}\Omega$ (a reference configuration $^{t}\Omega$, a virtual intermediate configuration $^{i}\Omega$, and a current configuration $^{t+\Delta t}\Omega$) is considered in two steps, the motion of the body from $^{t}\Omega$ to $^{i}\Omega$ and then from $^{i}\Omega$ to $^{t+\Delta t}\Omega$, as shown in Figure 2. The motion from $^{t}\Omega$ to $^{i}\Omega$ is purely plastic and irreversible. Therefore, the configuration $\Omega^{i}$ can be considered an unstressed configuration. The motion from $^{i}\Omega$ to $^{t+\Delta t}\Omega$ is purely elastic and reversible. The deformation gradient for the motion from $^{t}\Omega$ to $\Omega^{i}$ is denoted by $\mathbf{F}^{p}$, and the deformation gradient for the motion from $^{i}\Omega$ to $^{t+\Delta t}\Omega$ is denoted by $\mathbf{F}^{e}$ in matrix form. When the motion from $^{t}\Omega$ to $^{t+\Delta t}\Omega$ is continuous, the deformation gradient has the following non-cumulative representation in its plastic and elastic parts [12,13]:

$$\mathbf{F} = \frac{\partial \mathbf{x}}{\partial \bar{\mathbf{x}}} \frac{\partial \bar{\mathbf{x}}}{\partial \mathbf{X}} = \mathbf{F}^{e}\mathbf{F}^{p} \tag{11}$$

where $\mathbf{x}$ is the spatial coordinate, $\mathbf{X}$ is the material coordinate, and $\bar{\mathbf{x}}$ is the intermediate coordinate. Since the deformation measures are not linearly expressed in terms of displacements, the elastic and plastic components are not summable. Choosing a representation of the elastic part of deformation independent of rigid body motion (rotation), the deformation rate tensor $\mathbf{D}$ is given by the symmetric part of the velocity gradient $\mathbf{L}$ [13,14].

$$\mathbf{L} = \dot{\mathbf{F}}^{\mathbf{e}} \cdot (\mathbf{F}^{\mathbf{e}})^{-1} + \mathbf{F}^{\mathbf{e}}\dot{\mathbf{F}}^{\mathbf{p}}(\mathbf{F}^{\mathbf{p}})^{-1}(\mathbf{F}^{\mathbf{e}})^{-1}\text{and}$$
$$\mathbf{D} = \mathbf{D}^{e} + \left(\mathbf{F}^{e}\mathbf{D}^{p}\mathbf{F}^{e-1}\right)_{s} + \left(\mathbf{F}^{e}\mathbf{W}^{p}\mathbf{F}^{e-1}\right)_{s} \tag{12}$$

where $\mathbf{L} = \mathbf{D} + \mathbf{W}$

where $\mathbf{W}$ is the skew-symmetric part of the velocity gradient. The subscript s indicates the symmetric part. If the elastic components of the total strain are assumed to be minor, which is true for most soils, $\mathbf{F}^{e} \approx 1$ and the last term in Equation (12) is also small. Then, the rate of deformation reduces to the additive decomposition, as shown in Equation (13).

$$\mathbf{D} = \mathbf{D}^{e} + \mathbf{D}^{p} \tag{13}$$

where $\mathbf{D}^{e}$ and $\mathbf{D}^{p}$ are the elastic and plastic parts of the total strain rate, respectively.

The corotational form of the stress–strain relationship for the elastoplastic material was expressed in the following form (Equation (14)):

$$\dot{\sigma}^{\nabla} = \mathbf{C}^{ep} : \mathbf{D} \tag{14}$$

where $\mathbf{C}^{ep}$ is the tangential elastoplastic stiffness tensor, which may be a function of the current state of the net stress, matric suction, strains, and some other internal variables.

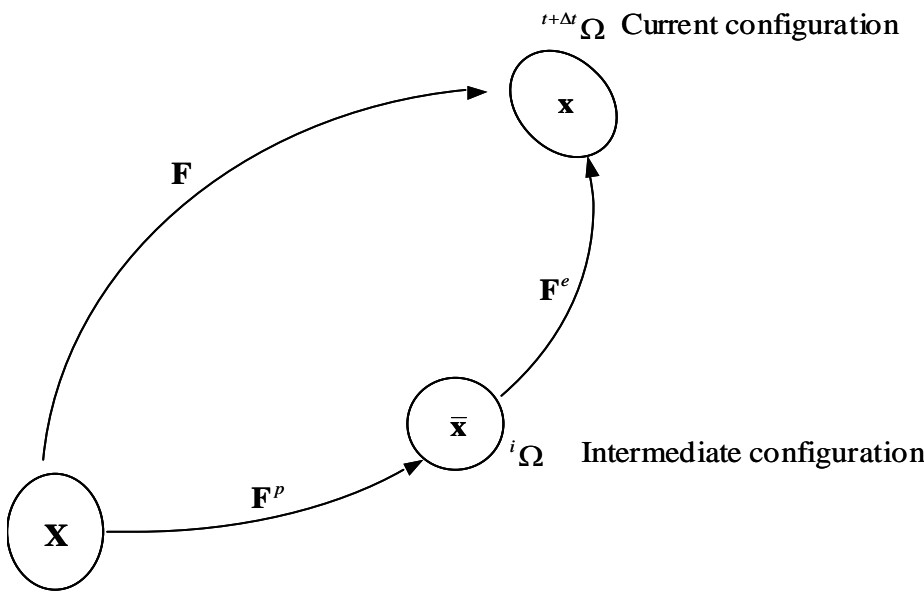

**Figure 2.** Multiplicative decomposition of elastic–plastic deformations.

Among the many forms of objective stress rates, the Green–Naghdi stress rate was used in this study, as shown in Equation (15).

$$\boldsymbol{\sigma}^{\nabla} = \frac{D}{Dt}\boldsymbol{\sigma} - \boldsymbol{\Omega} \cdot \boldsymbol{\sigma} - \boldsymbol{\sigma} \cdot \boldsymbol{\Omega}^{T} \tag{15}$$

where the angular velocity is given by $\boldsymbol{\Omega} = \dot{\mathbf{R}} \cdot \mathbf{R}^{T}$

The rotation tensor $\mathbf{R}$ was calculated using the polar decomposition theorem, which states that any deformation gradient tensor $\mathbf{F}$ can be multiplicatively decomposed into a product of an orthogonal matrix $\mathbf{R}$ and a symmetric tensor $\mathbf{U}$, called the right stretch tensor. The deformation gradient in the index notation is given by $F_{ij} = \frac{\partial x_i}{\partial X_j} = R_{ik}U_{kj}$.

By rearranging the objective rates and applying the net stress principle to the total objective rate, the net stress is expressed as:

$$\frac{D\sigma_{ij}}{Dt} = \sigma_{ij}^{n\nabla} - \delta_{ij}\dot{p}^{g} + \Omega_{ik}\sigma_{kj} + \sigma_{ik}\Omega_{kj}^{T} \text{ and } \frac{D\sigma_{ij}}{Dt} = C_{ijkl}^{ep}D_{kl} - \delta_{ij}\dot{p}^{g} + \Omega_{ik}\sigma_{kj} + \sigma_{ik}\Omega_{kj}^{T} \tag{16}$$

### *3.3. Stress-State Variables for Unsaturated Soils*

The net stress and matric suction are widely accepted as the stress-state variables to define the mechanical behavior of unsaturated soils [1] and were also used in this study. For large deformation analysis, an objective stress measure must be used to consider the effect of rotation. Therefore, the objectivity of the net stress must be derived before using appropriate objective stress measures to the net stress. The net stress is given by

$$\boldsymbol{\sigma} = \boldsymbol{\sigma}^{n} - p^{g}\mathbf{I} \tag{17}$$

where $\boldsymbol{\sigma}$ is the total stress tensor, $\boldsymbol{\sigma}^{n}$ is the net stress tensor, and $p^{g}$ is the pore air pressure. The conventional solid mechanics sign convention is used in the above equation. Since the net stress is defined in terms of the Cauchy stress tensor, which is not an objective measure of stress, a suitable rate form was established for the net stress equation. Equation (18) was derived by taking the time derivative of Equation (17).

$$\frac{D}{Dt}\boldsymbol{\sigma} = \frac{D}{Dt}\boldsymbol{\sigma}^{n} - \frac{D}{Dt}p^{g}\mathbf{I} \tag{18}$$

where $\frac{D}{Dt}$ indicates the material time derivative.

The net stress equation was rewritten in corotational form, as shown in Equation (19).

$$\boldsymbol{\sigma}^{\nabla} = \boldsymbol{\sigma}^{n\nabla} - \dot{p}^{g}\mathbf{I} \tag{19}$$

where $\boldsymbol{\sigma}^{\nabla}$ is the objective total stress rate, $\boldsymbol{\sigma}^{n\nabla}$ is the objective net stress rate, and $\dot{p}^{g}$ is the pore air pressure. The objective form of the net stress equation was incorporated into the virtual work equation.

*3.4. Boundary Conditions*

Solid, water, and air displacements; solid traction; and pore pressure boundary conditions were considered in the derivation. These boundary conditions are specified on different portions of the boundary surface $^{t+\Delta t}S$ of the soil body at a generic time $t + \Delta t$ and are defined as follows:

- Solid displacement boundary condition:

  $^{t+\Delta t}u_i = {}^{t+\Delta t}\overline{u}_i$ on $^{t+\Delta t}S_u$

  where $^{t+\nabla t}\overline{u}_i$ is the specified value of solid displacement on the boundary surface $^{t+\Delta t}S_u$ at time $t + \Delta t$.

- Water displacement boundary condition:

  $^{t+\Delta t}U_i^l = {}^{t+\Delta t}\overline{U}_i^l$ on $^{t+\Delta t}S_{U^l}$

  where $^{t+\nabla t}\overline{U}_i$ is the specified value of water displacement on the boundary surface $^{t+\Delta t}S_U$ at time $t + \Delta t$.

- Air displacement boundary condition:

  $^{t+\Delta t}U_i = {}^{t+\Delta t}\overline{U}_i$ on $^{t+\Delta t}S_{U^g}$

  where $^{t+\nabla t}\overline{U}_i^g$ is the specified value of air displacement on the boundary surface $^{t+\Delta t}S_{U^g}$ at time $t + \Delta t$.

- Traction boundary condition:

  $^{t+\Delta t}\sigma_{ij}n_j = {}^{t+\Delta t}f_i^t$ on $^{t+\Delta t}S_T$

  where $^{t+\nabla t}f_i^t$ is the specified value of traction on the boundary surface $^{t+\Delta t}S_T$ at time $t + \Delta t$, $n_j$ is the unit normal, and $^{t+\Delta t}\sigma_{ij}$ is the total Cauchy stress tensor acting on the neighborhood of $^{t+\Delta t}S_T$.

- Pore water pressure boundary condition:

  $^{t+\nabla t}p^l = {}^{t+\nabla t}\overline{p}^l$ on $^{t+\nabla t}S_{p^l}$

  where $^{t+\nabla t}\overline{p}^l$ is the specified value of the pore water pressure at time $t + \Delta t$.

- Pore air pressure boundary condition:

  $^{t+\nabla t}p^g = {}^{t+\nabla t}\overline{p}^g$ on $^{t+\nabla t}S_{p^g}$

  where $^{t+\nabla t}\overline{p}^g$ is the specified value of the pore air pressure at time $t + \Delta t$.

*3.5. Incremental Equations and Newton's Method*

By knowing the equilibrium state of the soil body at time $t$, the state $\boldsymbol{\Pi}$ was defined by the known stresses, tractions, deformations, and history of the soil body. Let the right and left sides of the virtual work equation in the reference configuration be $I(\boldsymbol{\Pi})$ and $E(\boldsymbol{\Pi})$, respectively. At time $t + \Delta t$, a new equilibrium state was established for the body. Let $\Delta\boldsymbol{\Pi}$ be the change in state, which is the solution of $[I(\boldsymbol{\Pi} + \Delta\boldsymbol{\Pi}) - E(\boldsymbol{\Pi} + \Delta\boldsymbol{\Pi})] \cdot \delta\mathbf{v} = 0$. Denoting $\overline{\boldsymbol{\Pi}}$ as a guess for the new equilibrium state, the above equation can be expanded around the new guessed state, as follows:

$$[I(\mathbf{\Pi} + \Delta\mathbf{\Pi}) - E(\mathbf{\Pi} + \Delta\mathbf{\Pi})] \cdot \delta\mathbf{v} = \left[ I\left(\overline{\mathbf{\Pi}}\right) - E\left(\overline{\mathbf{\Pi}}\right) \right] \cdot \delta\mathbf{v} + \left[ \frac{\partial I\left(\overline{\mathbf{\Pi}}\right)}{\partial \overline{\mathbf{\Pi}}} \partial\overline{\mathbf{\Pi}} - \frac{\partial E\left(\overline{\mathbf{\Pi}}\right)}{\partial \overline{\mathbf{\Pi}}} \partial\overline{\mathbf{\Pi}} \right] \cdot \delta\mathbf{v} + \cdots \tag{20}$$

where $\partial\overline{\mathbf{\Pi}}$ is the increment between the correct equilibrium state $\mathbf{\Pi} + \Delta\mathbf{\Pi}$ and the guessed equilibrium state $\overline{\mathbf{\Pi}}$ and $\delta\mathbf{v}$ is the virtual velocity. By taking first-order approximation, the above equation reduces to Equation (21), as follows:

$$\left[ \frac{\partial I\left(\overline{\mathbf{\Pi}}\right)}{\partial \overline{\mathbf{\Pi}}} \partial\overline{\mathbf{\Pi}} - \frac{\partial E\left(\overline{\mathbf{\Pi}}\right)}{\partial \overline{\mathbf{\Pi}}} \partial\overline{\mathbf{\Pi}} \right] \cdot \delta\mathbf{v} = -\left[ I\left(\overline{\mathbf{\Pi}}\right) - E\left(\overline{\mathbf{\Pi}}\right) \right] \cdot \delta\mathbf{v} \tag{21}$$

The successive solution of the above equation for various trial states $\overline{\mathbf{\Pi}}$ can be found until the right side of the equation becomes zero, i.e., $\overline{\mathbf{\Pi}}$ equal $\mathbf{\Pi} + \Delta\mathbf{\Pi}$ and the equilibrium is satisfied.

By substituting the stress measures and replacing the rate form with the incremental form, we obtain the following equations:

$$\dot{S}_{ij} = JF_{ik}^{-1}\left( C_{klmn}^{ep}D_{mn} + \Omega_{km}\sigma_{ml} + \sigma_{km}\Omega_{ml}^{T} \right)F_{lj}^{-T} \text{ and } \partial S_{ij} = JF_{ik}^{-1}\left( C_{klmn}^{ep}\partial D_{mn} + \partial\Omega_{km}\sigma_{ml} + \sigma_{km}\partial\Omega_{ml}^{T} \right)F_{lj}^{-T} \tag{22}$$

Equation (22) was substituted into Equation (10), and the incremental internal virtual work equation was derived, as shown in Equation (23).

$$\begin{aligned}
\partial^{t+\Delta t}W^{\text{int}} &= \int_{{}^{t}\Omega} \left( JF_{ik}^{-1}\left( C_{klmn}^{ep}\partial D_{mn} + \partial\Omega_{km}\sigma_{ml} + \sigma_{km}\partial\Omega_{ml}^{T} \right)F_{lj}^{-T} \right)\delta^{t+\Delta t}\varepsilon_{ij}{}^{t}d\Omega \\
&- \int_{{}^{t}\Omega} {}_{t}^{t+\Delta t}h_{ij}\delta_{t}^{t+\Delta t}\varepsilon_{ij}{}^{t}d\Omega
\end{aligned} \tag{23}$$

Equation (23) was simplified by choosing the current configuration to coincide with the reference configuration. This led to the deformation gradient becoming the identity tensor, while all the stress measures remained the same. This choice of the reference configuration is called the updated Lagrangian method [13–15]. This method is straightforward to use in computer codes because it requires only the coordinates of the body to be updated after each iteration so that the current configuration is also the reference configuration.

## 4. Finite Element Forms of Complete and Reduced Governing Equations

Closed-form solutions to the partial differential equations for many real-world problems do not exist. Therefore, a numerical technique, such as the finite element method, must be used to find approximate solutions for the system of equations. The finite element method is a powerful and widely used numerical method in geotechnical engineering to solve complex partial differential equations. To investigate the influence of relative accelerations and velocities of pore fluids on the overall dynamic behavior of unsaturated soils, two different formulations were derived in this study. The system of equations that considers the relative accelerations and velocities is called the *complete formulation*. In contrast, the system of equations simplified by neglecting the relative accelerations and velocities of the pore fluids is called the *reduced formulation*. The predictions from these two equations are compared in later sections.

### 4.1. Complete Formulation

The system of equations describing the dynamics of unsaturated soils, as listed below, consists of five equations and five unknowns.

$$n^s \rho^s \ddot{u}_j + n^l \rho^l \ddot{U}_j^l + n^g \rho^g \ddot{U}_j^g - \sigma_{ij,i} - \rho b_j = 0 \tag{24a}$$

$$\rho^l \ddot{U}_j^l - \left( \hat{k}_{ij}^l n^l \right) \dot{u}_i + \left( \hat{k}_{ij}^l n^l \right) \dot{U}_i^l + \left( \delta_{ij} p^l \right)_{,i} - \rho^l b_j = 0 \tag{24b}$$

$$\rho^g \ddot{U}_j^g - \left( \hat{k}_{ij}^g n^g \right) \dot{u}_i + \left( \hat{k}_{ij}^g n^g \right) \dot{U}_i^g + \left( \delta_{ij} p^g \right)_{,i} - \rho^g b_j = 0 \tag{24c}$$

$$\left( \frac{\partial n^l}{\partial \varepsilon_v} \right) \dot{u}_{i,i} + n^l \dot{U}_{i,i}^l + \left( \frac{n^l}{\Gamma^l} - \frac{\partial n^l}{\partial S} \right) \dot{p}^l + \left( \frac{\partial n^l}{\partial S} \right) \dot{p}^g = 0 \tag{24d}$$

$$\left( 1 - n - \frac{\partial n^l}{\partial \varepsilon_v} \right) \dot{u}_{i,i} + n^g \dot{U}_{i,i}^g + \left( \frac{\partial n^l}{\partial S} \right) \dot{p}^l + \left( \frac{n^g}{\Gamma^g} - \frac{\partial n^l}{\partial S} \right) \dot{p}^g = 0 \tag{24e}$$

By eliminating the water and air pressures in the momentum balance equations using the mass balance equations, the final set of equations (Equations (25a)–(25c)) was derived in terms of the displacement, velocity, and acceleration of solid, water, and air phases. In the finite element solution, the displacement fields of all three phases were considered the primary nodal unknowns, as shown in Figure 3a.

$$n^s \rho^s \ddot{u}_j + n^l \rho^l \ddot{U}_j^l + n^g \rho^g \ddot{U}_j^g - \sigma_{ij,i} - \rho b_j = 0 \tag{25a}$$

$$\rho^l \ddot{U}_j^l - \left( \hat{k}_{ij}^l n^l \right) \dot{u}_i + \left( \hat{k}_{ij}^l n^l \right) \dot{U}_i^l + \mu^{11} u_{i,ij} + \mu^{12} U_{i,ij}^l + \mu^{13} U_{i,ij}^g - \rho^l b_j = 0 \tag{25b}$$

$$\rho^g \ddot{U}_j^g - \left( \hat{k}_{ij}^g n^g \right) \dot{u}_i + \left( \hat{k}_{ij}^g n^g \right) \dot{U}_i^g + \mu^{21} u_{i,ij} + \mu^{22} U_{i,ij}^l + \mu^{23} U_{i,ij}^g - \rho^g b_j = 0 \tag{25c}$$

where
$$\mu^{11} = \left( \frac{a_{12} b_{21}}{(a_{22} a_{11} - a_{12} a_{21})} - \frac{a_{22} b_{11}}{(a_{22} a_{11} - a_{12} a_{21})} \right)$$
$$\mu^{12} = -\left( \frac{a_{22} b_{12}}{(a_{22} a_{11} - a_{12} a_{21})} \right)$$
$$\mu^{13} = \left( \frac{a_{12} b_{22}}{(a_{22} a_{11} - a_{12} a_{21})} \right)$$
$$\mu^{21} = \left( \frac{a_{21} b_{11}}{(a_{22} a_{11} - a_{12} a_{21})} - \frac{a_{11} b_{21}}{(a_{22} a_{11} - a_{12} a_{21})} \right)$$
$$\mu^{22} = \left( \frac{a_{21} b_{12}}{(a_{22} a_{11} - a_{12} a_{21})} \right)$$
$$\mu^{23} = -\left( \frac{a_{11} b_{22}}{(a_{22} a_{11} - a_{12} a_{21})} \right)$$
$$a_{11} = \left( \frac{n^l}{\Gamma^l} - \frac{\partial n^l}{\partial S} \right), a_{12} = \left( \frac{\partial n^l}{\partial S} \right), b_{11} = \left( \frac{\partial n^l}{\partial \varepsilon_v} \right), b_{12} = n^l$$
$$a_{21} = \left( \frac{\partial n^l}{\partial S} \right), a_{22} = \left( \frac{n^g}{\Gamma^g} - \frac{\partial n^l}{\partial S} \right), b_{21} = \left( 1 - n - \frac{\partial n^l}{\partial \varepsilon_v} \right) \text{ and } b_{22} = (n^g)$$
The water and air pressures can be calculated using

$$p^l = \mu^{11} u_{k,k} + \mu^{12} U_{k,k}^l + + \mu^{13} U_{k,k}^g \text{ and } p^g = \mu^{21} u_{k,k} + \mu^{22} U_{k,k}^l + \mu^{23} U_{k,k}^g \tag{26}$$

### 4.2. Simplified Formulation

The reduced formulation was derived by neglecting the pore fluids' relative accelerations and velocities. Such simplification results in three equations (the momentum balance equation for the mixture and the mass balance equations for the water and air phases). In this case, the momentum balance equation was solved considering the solid displacements as the primary nodal unknowns, as shown in Figure 3b, and the water pressure and air pressure were calculated using the mass balance equations. In this formulation, the changes in water and air pressures are directly related to the volumetric deformation of the solid

skeleton since the flow of fluids does not occur. This formulation simulates the undrained behavior of unsaturated soils.

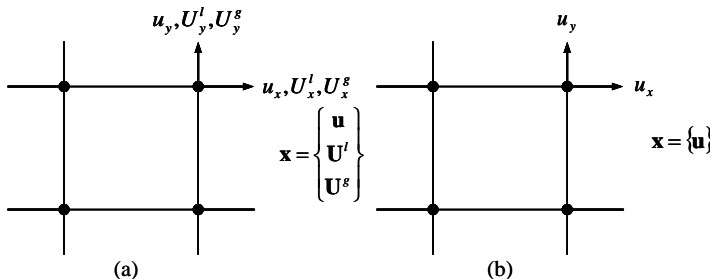

**Figure 3.** Nodal variables for the (**a**) u-U-U and (**b**) u formulations in 2 dimensions.

The final set of equations is summarized below (Equations (27)–(29)).

$$\rho \, \ddot{u}_j - \sigma_{ij,i} - \rho \, g_j = 0 \text{ in } {}^{t+\Delta t}\Omega \tag{27}$$

$$\left(n^l + \frac{\partial n^l}{\partial \varepsilon_v}\right) \dot{u}_{i,i} + \left(\frac{n^l}{\Gamma^l} - \frac{\partial n^l}{\partial p^c}\right) \dot{p}^l + \left(\frac{\partial n^l}{\partial p^c}\right) \dot{p}^g = 0 \text{ in } {}^{t+\Delta t}\Omega \tag{28}$$

$$\left(1 - n^l - \frac{\partial n^l}{\partial \varepsilon_v}\right) \dot{u}_{i,i} + \left(\frac{\partial n^l}{\partial p^c}\right) \dot{p}^l + \left(\frac{n^g}{\Gamma^g} - \frac{\partial n^l}{\partial p^c}\right) \dot{p}^g = 0 \text{ in } {}^{t+\Delta t}\Omega \tag{29}$$

*4.3. Finite Element Formulation*

The finite element formulations of the two formulations were derived using Galerkin's weighted residual method, considering isoparametric four-node quadrilateral elements. The time integration was performed using the Hilber–Hughes–Taylor-$\alpha$ method, which is a standard procedure in structural and geotechnical earthquake engineering. It allows for energy dissipation and second-order accuracy, which is not possible with the regular Newmark's method. The readers may refer to Wang and Chester [16], Wang et al. [17], and Ravichandran [8] for details on the derivation of weak form, shape functions, element-level residuals, and tangents.

## 5. Example Simulations

*5.1. Finite Element Model*

The finite element mesh for the unsaturated soil embankment used in this study is shown in Figure 4. The base of the embankment was assumed to be impermeable and fixed in both horizontal and vertical directions during the dynamic analyses. For the other three sides of the embankment, no displacement boundary conditions were applied so that they could move in any direction during shaking, and closed boundaries (no flow) were applied for the flow of water and air.

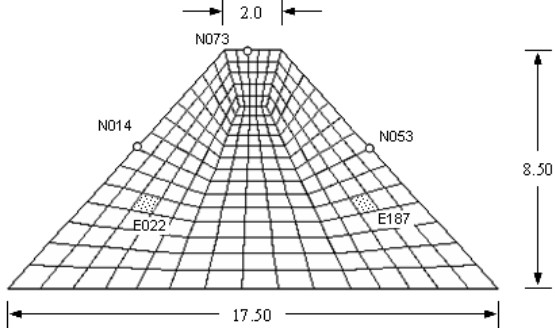

**Figure 4.** Finite element mesh and location of nodes and elements for records (all dimensions are in meters).

## 5.2. Constitutive models and model parameters

Stress–Strain Relationship of the Soil Skeleton

The stress–strain behavior of the solid skeleton was represented by an elastoplastic constitutive model based on the bounding surface concept. The schematic of the bounding surface model on stress-invariant space is shown in Figure 5. The original three-surface model for saturated cohesive soils [18] was modified for unsaturated soils by Muraleetharan and Nedunuri [19]. Additional parameters related to matric suction have been incorporated into the original model. This model uses the net stress $(\sigma_{ij} - p^g \delta_{ij})$ and matric suction $(S)$ as the stress-state variables. The modifications to the base model to incorporate the suction effects were based on the concepts proposed by Alonso et al. [20], Wheeler and Sivakumar [21], and Wheeler [22] for unsaturated soils. The simulations shown in this paper were carried out for an Oklahoma soil called Minco Silt (liquid limit = 28.0, plastic limit = 20.0, and USCS classification = CL). The model parameters for Minco Silt were obtained using the laboratory tests performed by Ananthanathan [23] and Vinayagam [24]. A complete list of the model parameters for Minco Silt and their values are listed in Table 1.

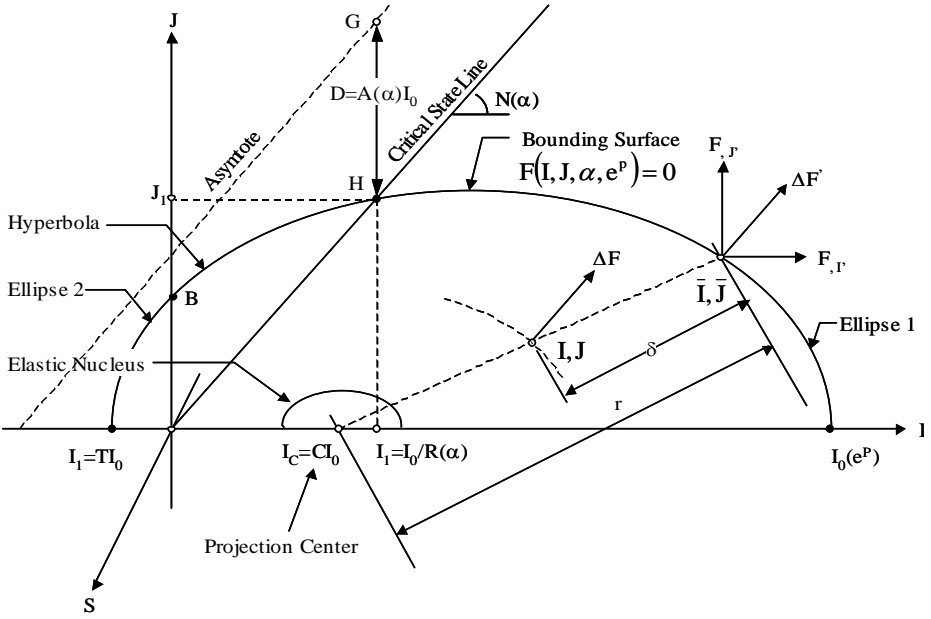

**Figure 5.** Schematic of a bounding-surface constitutive model in stress-invariant space.

**Table 1.** Calibrated bounding surface model parameters.

| Parameter | Value |
|---|---|
| Slope of virgin consolidation line in e-ln(p) plot, $\lambda$ | 0.0954 |
| Slope of swelling line in e-log(p) plot, $\kappa$ | 0.0103 |
| Slope of critical state line in compression, $M_c$ | 1.2678 |
| Ratio of extension to compression for slope of critical state line, $M_e/M_c$ | 1.00 |
| Elastic shear modulus, G $\left( \times 10^6 \, kPa \right)$ | 12.50 |
| Poisson ratio, $\nu$ | 0.20 |
| Mean net stress at which consolidation curve changes from linear in e-ln(p) space to linear in e-p space, $P_L$ | 33.80 |

**Table 1.** *Cont.*

| Parameter | Value |
|---|---|
| Shape parameter in compression corresponding to ellipse 1, $R_c$ | 4.20 |
| Ratio of extension to compression of shape parameter corresponding to ellipse 1, $R_e/R_c$ | 1.00 |
| Shape parameter in compression corresponding to hyperbola, $A_c$ | 0.05 |
| Ratio of extension to compression of shape parameter corresponding to hyperbola, $A_e/A_c$ | 1.00 |
| Shape parameter corresponding to ellipse 2, T | 0.01 |
| Projection center, C | 0.00 |
| Elastic zone parameter, $S_p$ | 1.00 |
| Positive model parameter, m | 0.02 |
| Degree of hardening in triaxial compression, $H_c$ | 0.80 |
| Ratio of extension to compression for hardening parameter, $H_e/H_c$ | 1.00 |
| Hardening parameter for states in immediate vicinity of I-axis, $H_o$ | 1.000 |
| Suction-dependent parameter 1, m | 4.703 |
| Suction-dependent parameter 2, N | 1.780 |
| Suction-dependent parameter 3, A | 0.420 |
| Suction-dependent parameter 4, B | 0.089 |

*5.3. Soil Water Characteristic Curve (SWCC) and Model Parameters*

Among the many SWCCs available in the literature, the model (Equation (30)) proposed by Brooks–Corey [25] was used in this study.

$$\begin{cases} \Theta = 1 & S < a \\ \Theta = \left(\frac{S}{a}\right)^{-n} & S > a \end{cases} \tag{30}$$

where $a$ and $n$ are the fitting parameters and $\Theta$ is the dimensionless water content given by $\Theta = \frac{\theta - \theta_r}{\theta_s - \theta_r}$. The subscripts $s$ and $r$ indicate the saturated and residual values of the volumetric water content, $\theta$, respectively. The SWCC parameters were determined by adjusting the model parameters until the model matched the experimental curve [23]. The model parameters are listed in Table 2. The relative permeability was calculated using the SWCC and the saturated permeability. The saturated permeability of $1.02 \times 10^{-8}$ m/s and the initial DOS of 0.43 were used in this analysis. The initial pore water and pore air pressures were $-30.0$ and $0.0$ kPa, respectively. The relative permeability of the water phase calculated using the van Genuchten method [25] was in the order of $10^{-3}$. This implies that the unsaturated soil permeability is $1.02 \times 10^{-11}$ m/s for the given initial conditions.

**Table 2.** Calibrated Brooks–Corey model parameters.

| Parameter | Value |
|---|---|
| Dry density (kN/m$^3$) | 14.14 |
| Parameter a | 5.5 |
| Parameter n | 0.5 |

*5.4. Static Analysis: Determination of the Initial Condition for Dynamic Analysis*

The initial stresses for dynamic analysis using an elastoplastic constitutive model were determined through static (gravity) analysis. The static analysis to compute the initial stresses was performed by choosing the time integration parameters as $\alpha = 0$, $\beta = 1.0$, and

$\gamma = 1.5$ [26,27]. The gravity load was increased, as shown in Figure 6, for the static analysis. Only the stresses were transferred to the dynamic analysis, and all the nodal and element fields were reset to zero at the end of the static analysis.

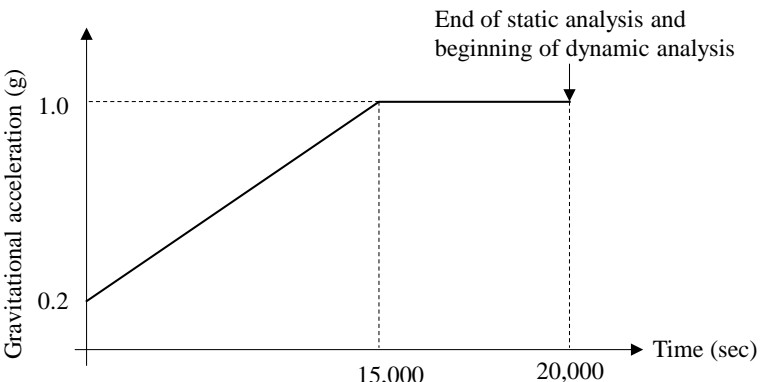

**Figure 6.** Gravity load–time history for static analysis.

### 5.5. Dynamic Analysis

For the dynamic analysis, the time integration parameters were changed to $\alpha = -0.3$, $\beta = 0.4225$, and $\gamma = 0.8$ [28] and the model embankment was shaken with the horizontal and vertical motions, as shown in Figure 7.

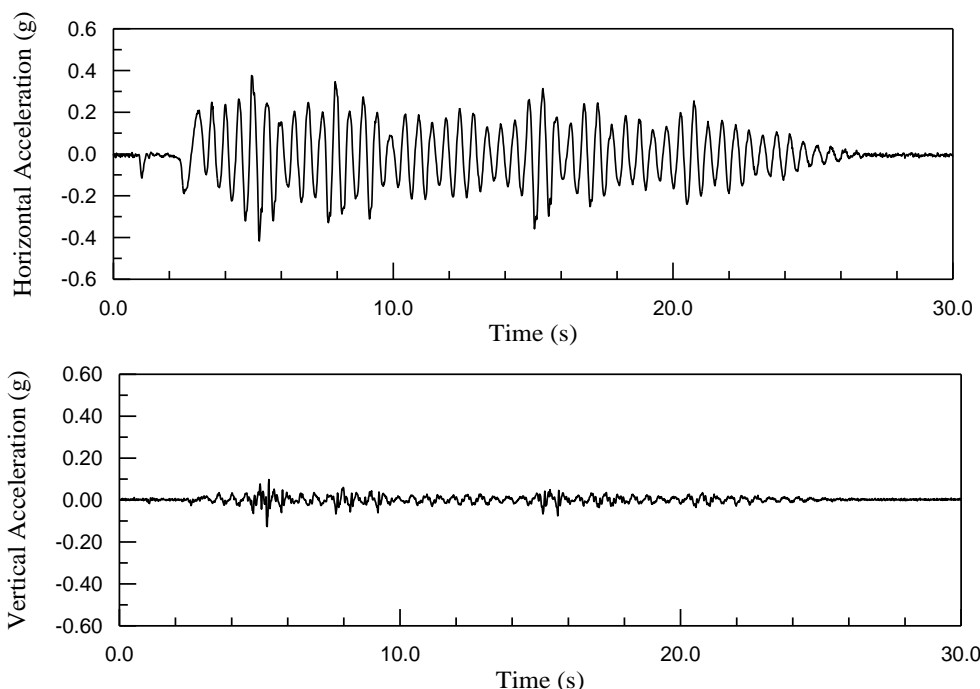

**Figure 7.** Input horizontal and vertical acceleration–time histories at the base of the model.

### 5.6. Comparison between Small and Large Deformation Analyses

The response of the unsaturated soil embankment discussed in the previous section was analyzed using both small and large deformation models using only the reduced formulation. The horizontal and vertical displacement–time histories at nodes N073 and N053 are shown in Figures 8 and 9, respectively. The time histories of the pore water pressure, pore air pressure, matric suction, and DOS in element E022 are shown in Figure 10. From these figures, it is clear that the model did not undergo significant deformation during the shaking. To increase the magnitude of the deformation, severe shaking was applied to the model by multiplying the amplitudes of the motions, as shown in Figure 7, by 6, and

then the results from the large and small deformation codes were compared. The horizontal and vertical displacements at nodes N073 and N053 are shown in Figures 11 and 12, respectively. The time histories of the pore water pressure, pore air pressure, matric suction, and DOS in element E022 are shown in Figure 13. We found that the large deformation analysis provides smaller vertical and horizontal displacements compared to the small deformation analysis. This is due to the initial stiffness contribution to the large deformation analysis at every time step coming from the updated Lagrangian formulation [15]. We also found that the large deformation analysis shows a minor change in pore water and air pressures. This response is consistent with the deformation response of the embankment.

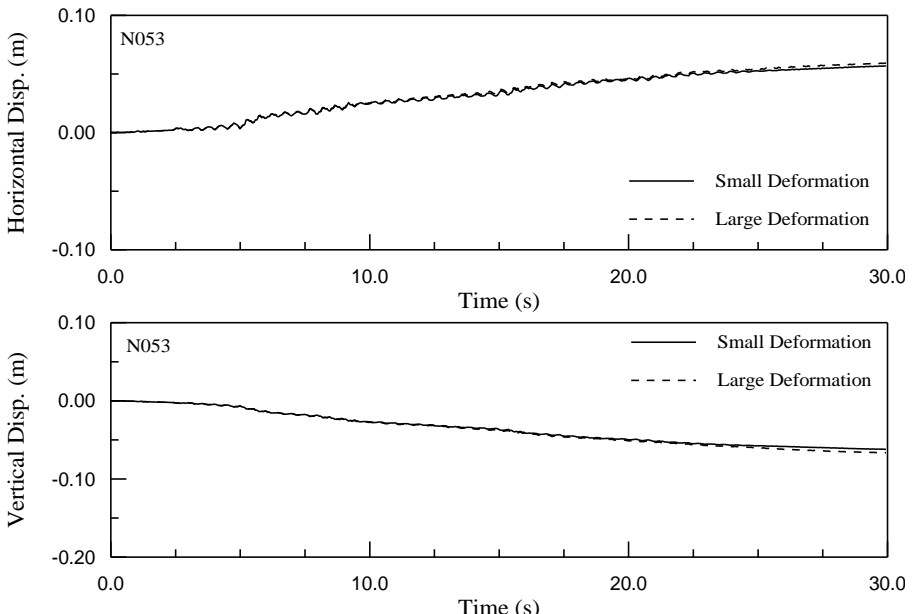

**Figure 8.** Horizontal and vertical displacement–time histories at node N053.

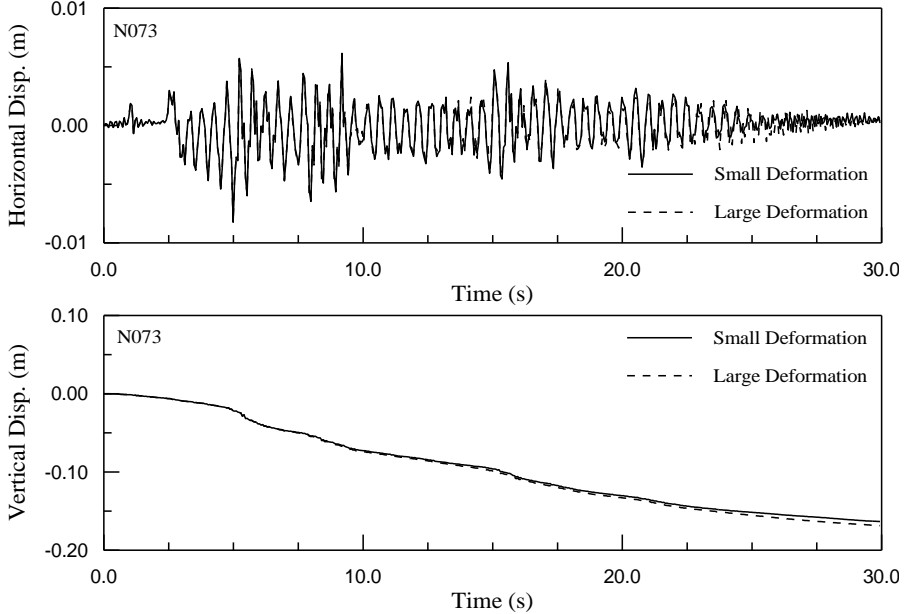

**Figure 9.** Horizontal and vertical displacement–time histories at node N073.

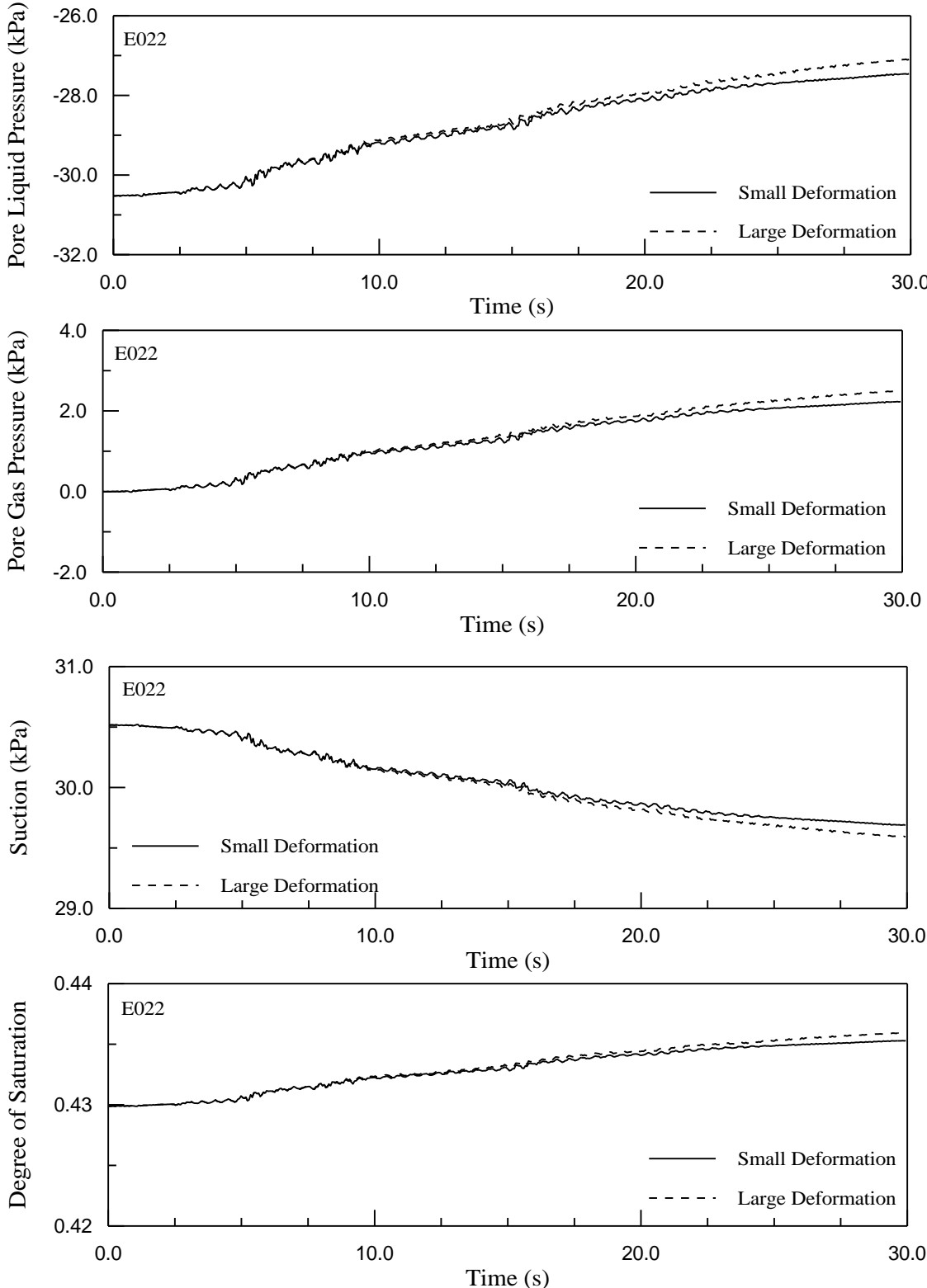

**Figure 10.** Fluid pressures, suction, and the degree of saturation–time histories at element E022.

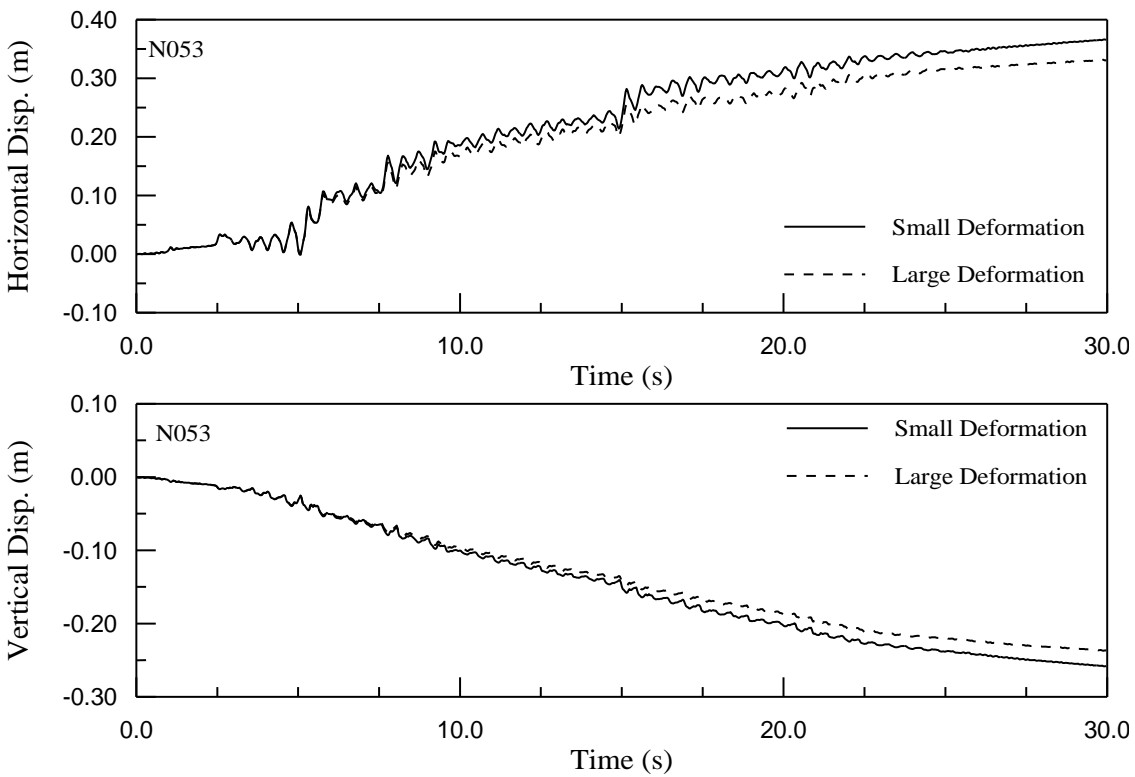

**Figure 11.** Horizontal and vertical displacement–time histories at node N053—severe shaking.

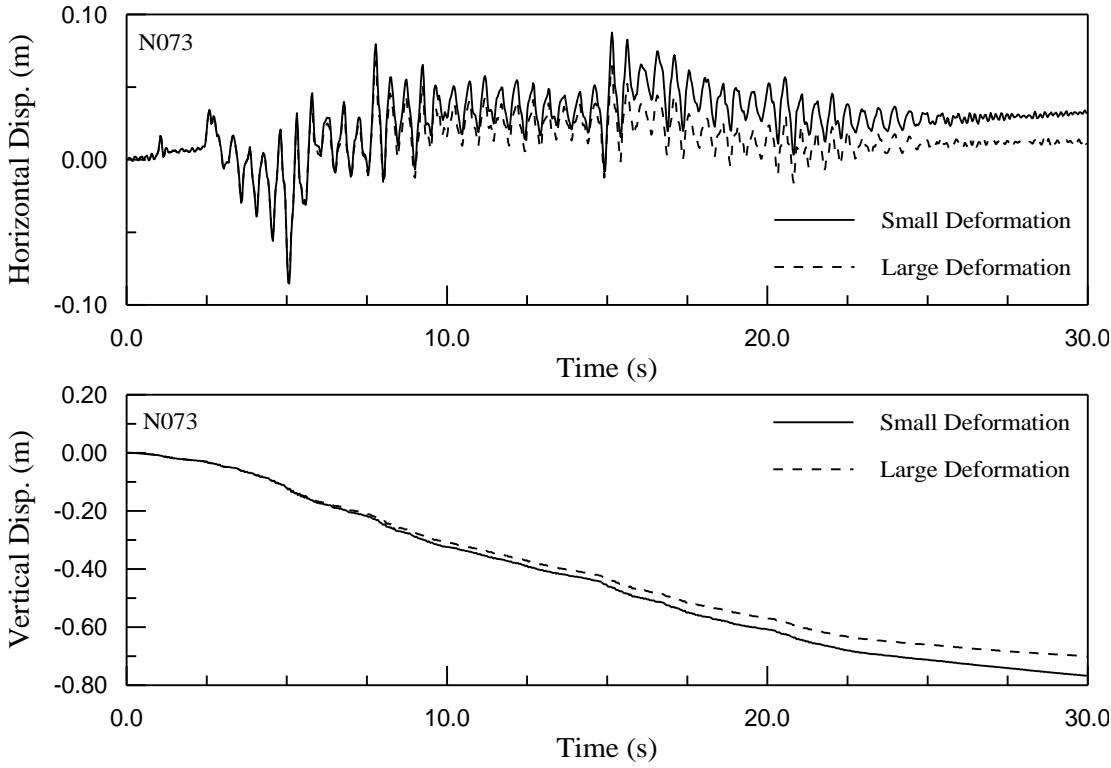

**Figure 12.** Horizontal and vertical displacement–time histories at node N073—severe shaking.

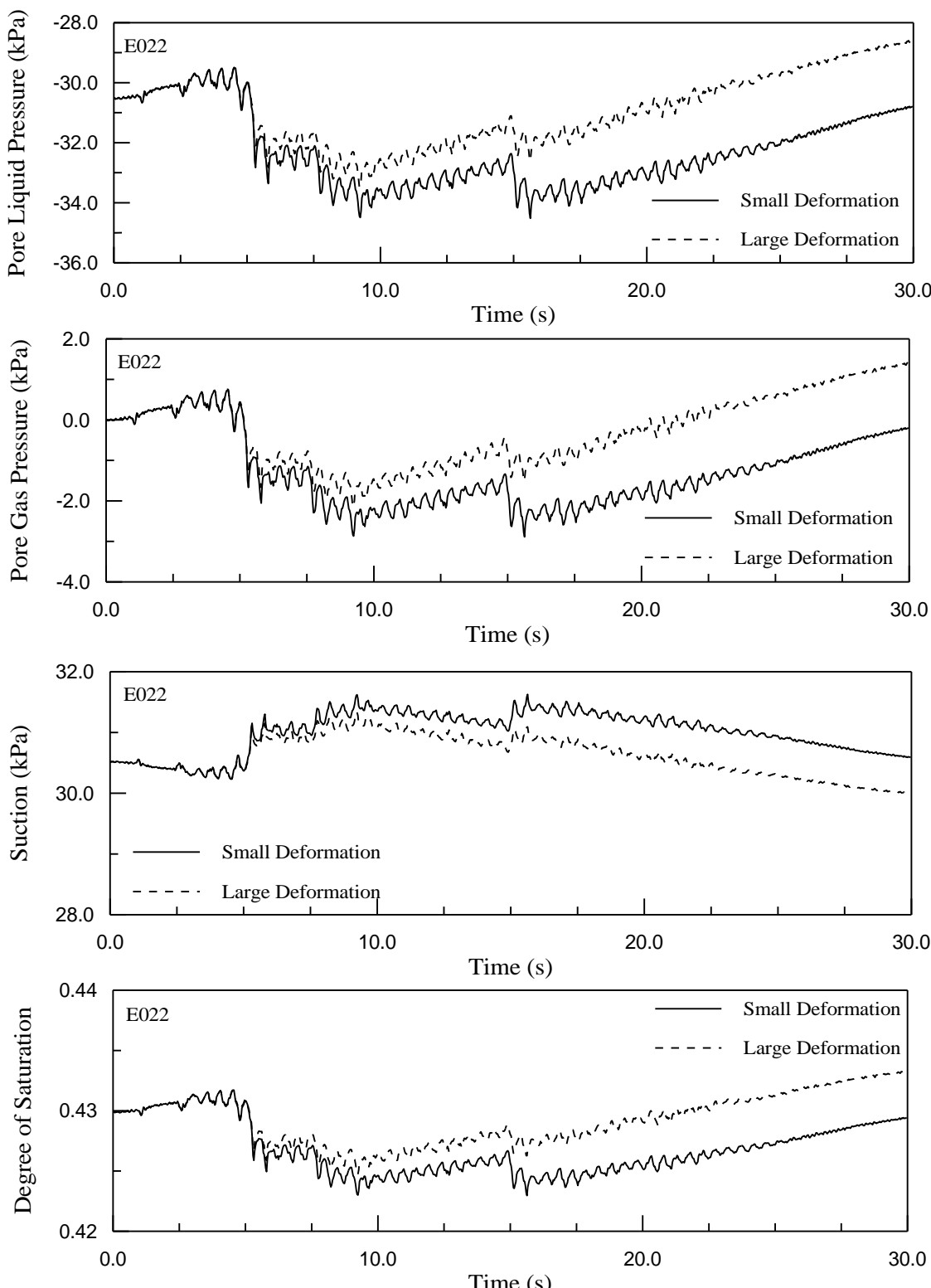

**Figure 13.** Fluid pressures, suction, and the degree of saturation–time histories in element E022–severe shaking.

*5.7. Comparison of Computational Efficiency of Complete and Reduced Formulations*

In the case of a four-node quadrilateral element for two-dimensional analysis, the complete formulation has 24-element degrees of freedom (DOF) and the reduced formulation has 8-element DOF. In the case of an eight-node brick element for three-dimensional

analysis, these numbers are 72 and 24 for the full formulation and the reduced formulation, respectively. These numbers provide an idea of the computational effort needed to solve these formulations. The embankment problem described earlier was run on an Intel-Xeon processor with a 3.0 GHz clock speed processor. The complete formulation took 1160 min, and the reduced formulation took 60 min to analyze 30 s of shaking using the same number of steps in both analyses. The average number of global iterations for the complete formulation was 5 and for the reduced formulation was 3. That is, in this example, the complete formulation required 20 times more computational time than that required by the reduced formulation.

## 6. Concluding Remarks

The following conclusions were made from this study:

- The large and small deformation theories for unsaturated soils were implemented within a finite element framework, and a large deformation analysis of the unsaturated soil was performed for the first time.
- The complete formulation for unsaturated soil was successfully solved for the first time. The effects of relative fluid accelerations and velocities on the overall behavior of unsaturated soils were estimated and compared. The reduced formulation is computationally efficient and numerically stable. It captures the overall behavior reasonably well for the soil considered in this study (Minco Silt) and can be used for the evaluation of earthquake effects on similar soils. However, caution should be exercised in using the reduced formulation for soils with larger permeability values.
- The small deformation analysis was observed to predict larger displacements compared to the large deformation analysis. Therefore, a large deformation analysis will likely produce an economical design of a geotechnical engineering structure.
- All predictions made by the computer code TeraDysac should be validated against experimental results. These validations are currently in progress.

**Author Contributions:** Writing—original draft, N.R.; writing—review and editing, T.V. All authors have read and agreed to the published version of the manuscript.

**Funding:** This research received no external funding.

**Data Availability Statement:** Data sharing not applicable. No new data were created or analyzed in this study. Data sharing is not applicable to this article.

**Conflicts of Interest:** The authors declare no conflict of interest.

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
