# Peer review of "Coupled Large Deformation Finite Element Formulations for the Dynamics of Unsaturated Soil and Their Application"

_geosciences, doi:10.3390/geosciences12090320_

Round 1

Reviewer 1 Report

Please see the attached PDF for comments. 

Author Response

Dear Reviewer,

Thank you for your time and effort in reviewing our paper and providing valuable comments. The quality of the revised paper has improved with your comments and questions. We have addressed all your comments and questions to the best of our knowledge. Please see below for your comments and responses. We have incorporated the responses in the revised manuscript as well.

Thanks,

Ravichandran on behalf of the authors.

Response to Reviewer #1

Major points:

  • The kinematics should always go first, e.g., the total deformation gradient F in Eq. 14 came after the Cauchy stress should be introduced earlier.

We have rearranged the sections in the revised manuscript.

  • The weak form, shape functions, element-level residuals, and tangents should be mentioned in Section 4. The authors are highly recommended to take (Wang and Chester, 2021; Wang et al., 2022) as references. It is not common to see zero contour plots in the numerical implementation manuscript. Thus, a contoured result should at least be demonstrated in the result section.

We have added a new section and cited the recommended references in the revised manuscript.

  • Code verification is needed to justify the correctness of the numerical implementation. Please take Wang and Hatami-Marbini (2021) as a reference.

The code was verified against hand calculations and another code (dysac) for 1, 2, and 4 elements at every step of the implementation process. Sample figures are given below.

4-Element verification: Vertical displacement (y-axis) vs. time (x-axis) at node 2.

2-Element verification: Vertical displacement (y-axis) vs. time (x-axis) at node 8.

Minor points:

  • In line 181, ∇σ should be called objective stress rate.

Corrected in the revised manuscript.

  • In line 198, one should use comma at the end of the sentence.

Corrected in the revised manuscript.

Reviewer 2 Report

The paper aims at presenting an unsaturated FEM formulation based on the large deformation theory, featuring a complete and a reduced form. An application case is proposed to show the potential of the formulation.

I have some major suggestions to improve the manuscript:

1) Stress the accent on the innovative character of the paper, to me it was not totally clear what is the actual innovation of this work. You should provide comparison with other formulations to help better contextualizing your work in the framework of past and current researches.

1) Describe in a more detailed manner spatial and temporal integration schemes of your formulation. This is a significant part of the implementation of the formulation totally missing.

2) Provide validation example/examples. I think it's a fundamental step, before applying the formulation to test it, and to provide robust validation examples, by comparing for example with analytical solutions.

3) Show more clearly (even through examples) the advantages of the large deformation formulation, what can provide more than the simplified approach (there is only a very short paragraph before the conclusions about this).

4) The application case is very poorly explained/documented. It is not clear if you are talking about an experimental case that you decided to reproduce with FEM or if it is a theoretical case.

5) Conclusion section is used improperly: you are introducing new concepts, that are not linked to the text. Conclusions should stress the accent on major finding of the presented study, the possible drawbacks and help the reader understanding the potential of extension of the research even to other fields.

I attach a pdf with some comments directly in the text in addition to these major points.

Author Response

Dear Reviewer,

Thank you for your time and effort in reviewing our paper and providing valuable comments. The quality of the revised paper has improved with your comments and questions. We have addressed all your comments and questions to the best of our knowledge. Please see below for your comments and responses. We have incorporated the responses in the revised manuscript as well.

Thanks,

Ravichandran on behalf of the authors.

Response to Reviewer #2

Comments were provided in a PDF as notes.

All the comments given in the PDF were addressed in the revised manuscript.

Other comments:

The paper aims at presenting an unsaturated FEM formulation based on the large deformation theory, featuring a complete and a reduced form. An application case is proposed to show the potential of the formulation.

I have some major suggestions to improve the manuscript:

1) Stress the accent on the innovative character of the paper, to me it was not totally clear what is the actual innovation of this work. You should provide comparison with other formulations to help better contextualizing your work in the framework of past and current researches.

The contribution of this paper is the development of a large deformation formulation for unsaturated soils (three phase materials) and comparing it with a widely used small deformation formulation. Developing large deformation finite element formulation for unsaturated soil by applying the two stress state effective stress concept has not been done. All the formulations found in the literature are for single-phase or two-phase materials.

1) Describe in a more detailed manner spatial and temporal integration schemes of your formulation. This is a significant part of the implementation of the formulation totally missing.

Deriving the governing equation and decomposition of strain and/or deformation gradient is the critical step for developing finite element formulation. The spatial and temporal decompositions are standard procedures. For example, the spatial decomposition was performed using Galerkin’s weighted residual method considering an isoparametric 4-node quadrilateral element. The time integration was performed using Hilber-Hughes-Taylor-Aphal method which is also a standard procedure in structural and geotechnical earthquake engineering. These are the reasons for not including the details in the paper.

2) Provide validation example/examples. I think it's a fundamental step, before applying the formulation to test it, and to provide robust validation examples, by comparing for example with analytical solutions.

The code for verified comparing the results from this code using hand calculations. A sample figure is provided in response to Reviewer #1. However, the validation was not performed as part of this study. We clearly mentioned this limitation in conclusion. We are building capabilities to conduct experiments and verify the code.

3) Show more clearly (even through examples) the advantages of the large deformation formulation, what can provide more than the simplified approach (there is only a very short paragraph before the conclusions about this).

The difference between simplified and full formulations is the consideration of the flow of fluids. The simplified formulation is an undrained formulation that can be used to simulate fine-grained soils (low permeable materials) under dynamic load (short duration). However, the full formulation considers the flow of both air and water and can be applied to any soil under any loading scenarios.

The difference between the large and small deformation formulations is the way the energy of the system or deformation is computed.  The rotation of an element during the large deformation is accurately considered, and the strain energy is computed accurately. This is not possible in a small deformation formulation where additive decomposition is used. Large deformation formulation is very useful to model soil-pile interaction under lateral load, slope failures, and other geotechnical engineering problems which involve large deformation.

4) The application case is very poorly explained/documented. It is not clear if you are talking about an experimental case that you decided to reproduce with FEM or if it is a theoretical case.

It should be noted that there is no closed-form solution to real-world problems associated with the dynamics of unsaturated soils. The purpose of this study is to develop a finite element-based mathematic model and then software for understanding the dynamics of unsaturated soil systems. As you know, verified and validated finite element models can be used to gain further insights into complex system behaviors. We believe that the proposed model serves that purpose.

5) Conclusion section is used improperly: you are introducing new concepts, that are not linked to the text. Conclusions should stress the accent on major finding of the presented study, the possible drawbacks and help the reader understanding the potential of extension of the research even to other fields.

We have revised the conclusion to convey the message clearly.

Reviewer 3 Report

Manuscript number: Manuscript Geosciences-1804696

Title: A Coupled Large Deformation Finite Element Formulations for the Dynamics of Unsaturated Soil and Its Application.

Authors: Nadarajah Ravichandran , Tharshikka Vickneswaran

Summary: In the present paper, a finite element model developed for the dynamics of unsaturated soils, considering the interaction among the bulk phases and interfaces is presented to simulate the geomaterials in application with geotechnical engineering.

I have the following comments that the authors may need to consider in the revision.

(1) A lot of related studies have been done before. Authors should list the previous research results, review them and compare them with the methods of this paper to explain the shortcomings of previous studies.

(2) So many references were piled up in the introduction. The literature should be referenced and described according to the purpose of this study.

(3) How to evaluate the accuracy of numerical simulation?

(4) Are scaling variables used in this model? Do authors have any conditions to satisfy the mesh independence of FEM?

(5) Numerical study:

   a) How is the dynamic boundary of flow mode in your applications tested?

   b) The model of FEM in this paper is applied to small strain or finite strain or large strain? Please provide the range of this model for the scientist in this field. What is different types of mode?

   c) In Fgure 8 and 9, We don’t see the difference between small strain and large strain. The results is reseanable for tests. Why is that?

Author Response

Dear Reviewer,

Thank you for your time and effort in reviewing our paper and providing valuable comments. The quality of the revised paper has improved with your comments and questions. We have addressed all your comments and questions to the best of our knowledge. Please see below for your comments and responses. We have incorporated the responses in the revised manuscript as well.

Thanks,

Ravichandran on behalf of the authors.

Response to Reviewer #3

  • A lot of related studies have been done before. Authors should list the previous research results, review them and compare them with the methods of this paper to explain the shortcomings of previous studies.

We have discussed the previous studies in the revised manuscript.

  • So many references were piled up in the introduction. The literature should be referenced and described according to the purpose of this study.

The introduction has been revised to show only the related studies.

  • How to evaluate the accuracy of numerical simulation?

This is one of the shortcomings of this study which we have clearly mentioned in the discussion. However, we have verified the implementation with results from hand calculation and another code.

  • Are scaling variables used in this model? Do authors have any conditions to satisfy the mesh independence of FEM?

A parametric study was conducted by changing the number of elements, and the mesh that does not influence the computed results (displacement in this case) was chosen. Since the full formulation is numerically unstable and computationally expensive, we needed to choose a mesh that is accurate at the same time, computationally less expensive.

  • Numerical study:
    • How is the dynamic boundary of flow mode in your applications tested?

We used a closed boundary in this study since the simulations were conducted at a constant degree of saturation. If one wants to allow infiltration from the top and sides for a long period of time, then the boundaries must be opened numerically to allow water to seep out. However, such boundary implementation requires balancing the mass and continuity in the model, which is not implemented in the code at this time.

  • The model of FEM in this paper is applied to small strain or finite strain or large strain? Please provide the range of this model for the scientist in this field. What is different types of mode?

The model is applicable to small and large strain problems as long as the model is numerically stable. The rotation during large deformation is decoupled using an objective stress rate for accurate energy calculation. So, the model is capable of modeling small and large deformations.

  • In Fgure 8 and 9, We don’t see the difference between small strain and large strain. The results is reseanable for tests. Why is that?

The deformation experience by the model is small, and as expected, both small and large deformation theories must show close results. It is similar to the results from elastic and elastoplastic analyses when the system behaves within the elastic range. We are working on analyzing soil-structure interaction problems that induces large deformation. We clearly see the difference and the powerfulness of large deformation theory.

Round 2

Reviewer 1 Report

The author addressed all my comments; the manuscript could be published in its current form. 

Author Response

N.A.

Reviewer 2 Report

Major suggestion

1a) How about the works by Sanavia et al (2002), Gawin and Sanavia (2009)? They deal with unsaturated soil and large strain, may be pertinent to be added in the introduction?

1b) I noticed you added a sentence about space discretization, but nothing about time even if it was mentioned in the answer. Could you please add also that part in the main text, just a sentence would be sufficient for completeness and to help a generic reader of Geosciences.

2) When I talk about validation I don't mean verification,so I don't understand when you refer to a figure prepared for another reviewer. To me validation should be carried out considering for example an analytical solution or an experiment. I am aware that for large deformation there are no analytical solutions to validate your code, so it would be acceptable a small deformation or only flow problem to validate at least some parts of the formulation.

I understand this work is still under development, so I expect that in future publications the formulation will be complemented with validations, to make it more robust.

4) If I understood correctly the case presented is theoretical but based on reasonable assumptions about material parameters (Minco Silt) and loading conditions. Since you described a large deformation formulation I believe it would have been more pertinent an example where large deformations occur, and it is clearly visible that the large deformation formulation can reproduce them, hence the entire phenomenon under analysis, while the small deformation formulation only reproduce the phenomenon until  a certain point. For example, a slope failure fully reproduced by the first formulation and only until failure onset by the second formulation.

Moreover as suggested also by another reviewer, I also suggest to consider the idea of adding a figure showing at least a contour of strain or pressure resulting from the simulations.

Lastly I don't think all the boundary conditions are listed in Section 6, you just mention impermeability and fixity of the bottom edge, and the other edges?

Minor corrections

- In the abstract "In this paper, a set of fully coupled governing equations
IS developed for the dynamics of unsaturated soils, considering the interaction among the bulk phases and interfaces"

- At pg 15 "To increase the magnitude of the deformation, severe shaking was applied to the model by multiplying the amplitudes of the motions shown in Figure 8 by six"

Are you sure it is the quantity in figure 8 that you amplify?

- I think that section 5 should become a subsection of section 6. You clearly assign numerical values to parameters that are used to describe the behavior of the embankment of the example, so it shouldn't be part of the description of the formulations.

- Pag 20 when you mention TeraDysac can you provide a reference website or say if it is a internal research code/academic code

Author Response

Dear Reviewer#2,

Thank you for the opportunity to revise the paper one more time and improve its quality. We have addressed your comments to the best of our knowledge. Please see our responses below and the revised manuscript.

Thank you.

Ravi

1a) How about the works by Sanavia et al (2002), Gawin and Sanavia (2009)? They deal with unsaturated soil and large strain, may be pertinent to be added in the introduction?

Response: Thank you for pointing out two important papers. We have briefly mentioned these two papers in the introduction section.

1b) I noticed you added a sentence about space discretization, but nothing about time even if it was mentioned in the answer. Could you please add also that part in the main text, just a sentence would be sufficient for completeness and to help a generic reader of Geosciences.

Response: Thank you for your suggestions. We mentioned it in the response letter but not in the text. We have added this information in the revised manuscript.

2) When I talk about validation I don't mean verification,so I don't understand when you refer to a figure prepared for another reviewer. To me validation should be carried out considering for example an analytical solution or an experiment. I am aware that for large deformation there are no analytical solutions to validate your code, so it would be acceptable a small deformation or only flow problem to validate at least some parts of the formulation. I understand this work is still under development, so I expect that in future publications the formulation will be complemented with validations, to make it more robust.

Response: Thank you for your understanding. We are building an experimental set up in our outdoor lab and will provide validation for the large deformation formulation. We have clearly mentioned the future work /limitation of the study in the manuscript to warn the readers and look for future publications.

4) If I understood correctly the case presented is theoretical but based on reasonable assumptions about material parameters (Minco Silt) and loading conditions. Since you described a large deformation formulation I believe it would have been more pertinent an example where large deformations occur, and it is clearly visible that the large deformation formulation can reproduce them, hence the entire phenomenon under analysis, while the small deformation formulation only reproduce the phenomenon until  a certain point. For example, a slope failure fully reproduced by the first formulation and only until failure onset by the second formulation.

Response: This is a great suggestion. We are developing a paper on behavior of geotechnical systems under multiple hazards that involves temporal variation of factor of safety and deformation of slopes, axial and lateral deformation of deep foundations, and shallow foundations. Please look for a couple of papers in the near future.

Moreover as suggested also by another reviewer, I also suggest to consider the idea of adding a figure showing at least a contour of strain or pressure resulting from the simulations.

Thank you for your suggestions and we agree with you and the other reviewer. Unfortunately, we don’t have an easy way to plot the contours at this time. We are working on such capabilities including animation. We will add such results in the future publications.

Lastly I don't think all the boundary conditions are listed in Section 6, you just mention impermeability and fixity of the bottom edge, and the other edges?

Response: We have specified the boundary conditions for all the sides in the revised manuscript.

Minor corrections

- In the abstract "In this paper, a set of fully coupled governing equations
IS developed for the dynamics of unsaturated soils, considering the interaction among the bulk phases and interfaces"

Response: It is corrected in the revised manuscript.

- At pg 15 "To increase the magnitude of the deformation, severe shaking was applied to the model by multiplying the amplitudes of the motions shown in Figure 8 by six"

Are you sure it is the quantity in figure 8 that you amplify?

Response: It is Figure 7. We have corrected it in the revised manuscript.

- I think that section 5 should become a subsection of section 6. You clearly assign numerical values to parameters that are used to describe the behavior of the embankment of the example, so it shouldn't be part of the description of the formulations.

Response: We have moved Section 5 into Section 6 as a sub section. Thanks for your comments.

- Pag 20 when you mention TeraDysac can you provide a reference website or say if it is a internal research code/academic code.

Response: A reference is provided in the revised manuscript.

Reviewer 3 Report

The authors have addressed all my comments in the revised manuscript. This manuscript is accepted for publication.

Author Response

N.A.

Round 3

Reviewer 2 Report

The comments and requests of my last revision were all addressed. Therefore, I accept the paper in the present form

Thanks